# Morphological Ontogeny and Ecology of a Common Peatland Mite, *Nanhermannia coronata* (Acari, Oribatida, Nanhermanniidae)

**DOI:** 10.3390/ani13223590

**Published:** 2023-11-20

**Authors:** Stanisław Seniczak, Anna Seniczak

**Affiliations:** 1Department of Evolutionary Biology, Faculty of Biological Sciences, Kazimierz Wielki University, 85-093 Bydgoszcz, Poland; 2Faculty of Applied Ecology, Agricultural Sciences and Biotechnology, Inland Norway University of Applied Sciences, 2318 Elverum, Norway

**Keywords:** oribatid mites, juveniles, leg setation, stage structure, ecology, integrated taxonomy approach

## Abstract

**Simple Summary:**

*Nanhermannia coronata* is a common and abundant oribatid species in peatlands, but it can be easily mistaken for *N*. *sellnicki* as an adult. The identity of adults of *N*. *coronata* investigated herein from several sites in Norway and Ireland was supported by the COI sequence data. Based on this material, the morphological ontogeny of *N*. *coronata* was investigated, and some characters were found that clearly differentiate *N*. *coronata* from *N*. *sellnicki*, like the number of setae on femora of adults and tritonymphs, the shape of insertions of prodorsal seta *in* and all gastronotal and adanal setae of juveniles. Our ecological observations confirm a common occurrence of *N. coronata* in raised bogs, a high percentage of juvenile stages in populations and a preference of this species for humid microhabitats, whereas *N*. *sellnicki* is less common than *N*. *coronata* and occurs in drier habitats.

**Abstract:**

*Nanhermannia coronata* Berlese, 1913, is a common and abundant oribatid species in peatlands but can be easily mistaken for *N*. *sellnicki* Forsslund, 1958, as an adult. Therefore, the identity of adults of *N*. *coronata* from several sites in Norway and Ireland was supported by the COI sequence data, and based on this material, the morphological ontogeny of this species is described and illustrated to highlight the differences between *N*. *coronata* and *N*. *sellnicki*. In all juvenile stages of *N*. *coronata*, the bothridial seta is absent, but two pairs of exobothridial setae are present, including short *exp* and *exa* reduced to its alveolus. In the larva, seta *f*_1_ is setiform, but in the nymphs, it is reduced to its alveolus. Most prodorsal and gastronotal setae of larva are short, and of nymphs they are long. In all instars, the leg segments are oval in cross section and relatively thick, and many setae on tarsi are relatively short, thick and conical, except for longer apical setae. Seta *d* accompanies solenidion σ on all genua, φ_1_ on tibia I and φ on other tibiae. We found some morphological characters that clearly differentiate *N*. *coronata* from *N*. *sellnicki*, like the number of setae on femora of adults and tritonymphs, the shape of insertions of prodorsal seta *in* and all gastronotal and adanal setae of juveniles; in *N*. *sellnicki*, these setae are inserted in small individual depressions, whereas in *N. coronata*, these depressions are absent. Our ecological observations confirm a common occurrence of *N. coronata* in raised bogs, a high percentage of juvenile stages in its populations and a preference of this species for humid microhabitats, whereas *N*. *sellnicki* is less common than *N*. *coronata* and occurs in drier habitats.

## 1. Introduction

*Nanhermannia* Berlese, 1913, *sensu stricto* (*N*. *nanus* Nicolet, 1855, as the type species) comprises 31 nominative species [1]; all are of medium size as adults. The diagnosis of adults of this genus was given by Seniczak et al. [2] as follows: Adults are medium-sized (450–660 µm), elongated, brown to dark brown and narrow, with bothridia and bothridial setae situated on top of the prodorsum; there are protuberances on the posterior part of the prodorsum with sclerotized tubercles, often extending above the anterior part of the notogaster. The notogaster is cylindrical, with a coarse structure of pits and 15 pairs of long setae, curved and appressed to body, extended ventrally and connected ventro-medially; an arched suture delimits the genital and aggenital area laterally (diagastry). The aggenital plate is fused with the epimere, and the adanal plate is fused with the notogaster, but still recognizable; the formula of the epimeral setae is usually 3-1-3-3 or 3-1-3-4, but in some species, hypertrichy on some epimeres occurs; 7–10 pairs of genital setae, two pairs of aggenital setae, two pairs of anal setae and three pairs of adanal setae are present. Legs have one claw.

Identification of *Nanhermannia* species is not easy because they are relatively similar to each other by having a similar body shape, structure of pits and shape of notogastral setae. The species differ from one another by the shape of protuberances on the posterior margin of the prodorsum and the number of sclerotized tubercles, which are considered diagnostic in *Nanhermannia*, but these characters vary in some species [3,4,5,6,7,8,9,10], including *N*. *coronata* Berlese, 1913, being a source of confusion. For example, during the revision of the twenty-year-old oribatid mite collection of the Institute of Biology, University of Latvia, a high discrepancy in the identification of *N*. *coronata* was detected, and ca. 50% of specimens (out of 40 studied) were wrongly identified [11]. Therefore, more investigations on the morphology of *Nanhermannia* species are required, including the juvenile stages and molecular investigations to improve the diagnosis of species of this genus, and in some cases to support the identity of individuals within species.

Systematic problems in *Nanhermannia* also occur. For example, in the past, *N. coronata* was confused with *N. nana* (Nicolet, 1855) *sensu* Willmann [12], which was clarified by Forsslund [13], and it was confirmed by Solhøy [14,15] that in Norwegian oligotrophic bogs, *N. coronata* was present. Another example is considering *N*. *coronata* by Subías [1] as a junior synonym of *N*. *dorsalis* (Banks, 1896), whereas Weigmann [8] and Norton and Ermilov [16] treated it as a separate species, and we agree with the latter opinion.

Identification of *Nanhermannia* species can also be problematic in ecological investigations. For example, *N*. *coronata* can be mistaken for *N*. *sellnicki* Forsslund, 1958. An identification of *N. coronata* is commonly based on the shape of protuberances on the posterior margin of the prodorsum and the number of sclerotized tubercles, which vary within this species, so the ecology of *N. coronata* given in some papers can be imprecise and needs improving. These species have undoubtedly different ecological preferences; *N*. *sellnicki* is less common than *N*. *coronata* and occurs in drier habitats [13], like birch forests, especially with understory formed by *Vaccinium* and *Empetrum*, while *N*. *coronata* is found in moist habitats, especially in raised bogs [15,17,18,19,20,21,22]. The latter species can be very abundant and dominant among the Oribatida [15,19,20,23] and among the mites [20]. For example, in western Norway, among nearly 60,000 mites collected from different peatland microhabitats, and represented by 154 species from all mite orders (Mesostigmata, Trombidiformes and Sarcoptiformes), *N*. *coronata* was the most abundant species (it made up 18% of all mite specimens) and occurred in about 90% of samples [20,22]. Such an abundant and common occurrence of *N*. *coronata* in peatlands requires better knowledge of the morphology of adults and juveniles, which justifies the need for the current study. The more so that in populations of this species, the juveniles are often very abundant, e.g., in peatlands in Norway, they constituted about 40% of individuals [20,22], so it is very important to include them in ecological analyses.

The morphology of juveniles of *Nanhermannia* is insufficiently known. According to Norton and Ermilov [15] and Seniczak et al. [2], the full morphological ontogeny of *N*. *comitalis* Berlese, 1916, *N*. cf. *coronata*, *N*. *nana* and *N*. *sellnicki* is known, which constitutes nearly 13% of all species. The morphological ontogeny of *N*. cf. *coronata* was investigated by Ermilov and Łochyńska [9], but these authors treated the leg setation generally, without labelling of leg setae, which we consider species-specific. Moreover, we found some differences in the morphology of juveniles of this species investigated by these authors and those studied herein, which probably illustrates the morphological variability of the species.

The aim of this paper is to describe the morphological ontogeny of *N*. *coronata* and compare it with that of congeners. The identity of specimens from several sites in Norway and Ireland is supported by the COI sequence data. We also add some data on the ecology of this species.

## 2. Material and Methods

### 2.1. Morphological and Biological Studies

The adults and juveniles of *N*. *coronata* used in morphological and biological studies were collected on 29 and 30 June 2020 (leg. A. Seniczak, K.I. Flatberg, K. Hassel and S. Roth) from an Atlantic raised bog located in Hitra (Hitra municipality, Trøndelag, Norway, 63°29′21.7” N, 8°52′25.1” E, 82 m a. s. l.). In total, 26 samples were collected. These samples were transported in plastic bags in cool boxes for four days to the laboratory of Bydgoszcz University of Science and Technology, Bydgoszcz, Poland, and extracted with Tullgren funnels for ten days into 90% ethanol. In these samples, *N*. *coronata* was the only member of *Nanhermannia*, and therefore, we considered the juveniles to belong to this species. The morphological ontogeny of *N*. *coronata* investigated herein is based on the abundant individuals from the transition zone between hummock and hollow in Hitra, but the morphological characters of instars were checked with those from other sites studied herein. We investigated the stage structure of mites, and based on 30 randomly selected adults, we determined the sex ratio, number of gravid females and carried eggs. We also measured the total body length (tip of the rostrum to the posterior edge of the notogaster) in the lateral aspect and the body width (widest part of the notogaster) in the dorsal aspect. In a similar way, we measured the morphological characters of juvenile and adult instars of *N*. *coronata* given in Table 1, as well as the size of anal and genital openings and setae perpendicularly to their length in μm. We used the microscopy Nikon Eclipse Ni.

The illustrations are limited to the body regions that show substantial differences between instars and were prepared from individuals mounted temporarily on slides in lactic acid. In the text and figures, we used the following abbreviations: rostral (*ro*), lamellar (*le*), interlamellar (*in*) and exobothridial (*exa*, *exp*) setae, bothridium (*bo*), bothridial seta (*bs*), notogastral or gastronotal setae (*c*-, *d*-, *e*-, *f*-, *h*-, *p*-series), cupules or lyrifissures (*ia*, *im*, *ip*, *ih*, *ips*, *iad*, *ian*), cheliceral seta (*cha*, *chb*), Trägårdh organ (*Tg*), palp setae (*sup*, *l*, *cm*, *acm*, *vt*, *ul*, *su*) and solenidion ω, epimeral setae (*1a*–*c*, *2a*, *3a*–*c*, *4a*–*d*), genital setae (*g*), adanal and anal setae (*ad*-, *an*-series), leg solenidia (σ, φ, ω), famulus (ε) and setae (*bv*, *ev, d*, *l*, *v*, *ft*, *tc*, *pv*, *a*, *s*, *p*, *u*). The leg setae *l* on femora, and *l* and *v* on tarsi were labelled according to their appearance in the ontogeny. The terminology used follows that of Grandjean [24,25,26,27,28] and Norton and Behan-Pelletier [29]. The species nomenclature follows partly Subías [1], Weigmann [8] and Norton and Ermilov [14].

For scanning electron microscopy (SEM), the mites were air-dried and coated with Au/Pd in a Polaron SC502 sputter coater and placed on Al-stubs with double-sided sticky carbon tape. Observations and micrographs were made with a QUANTA FEG 450 scanning electron microscope.

### 2.2. DNA Barcoding

For molecular studies, we used the specimens of *N*. *coronata* collected in raised bogs in southern, central and northern Norway, and Ireland (Table 1). In all locations, samples of *Sphagnum* mosses of 500 cm^3^ each were collected and extracted into 90% ethanol in the same way as described above. Additionally, we used publicly available DNA sequences of *N*. *coronata* from Finland and some other species of *Nanhermannia* identified (*N. nana* from Germany and Slovakia) and *N. comitalis* from Germany. We used species of putatively close genera as outgroups, *P*. *punctatus* (L. Koch, 1879) and *Camisia foveolata* Hammer, 1955.

Specimens of *N*. *coronata* from different locations were sent for DNA barcoding to the Canadian Centre for DNA Barcoding (CCDB) in Guelph, Canada. Each specimen was photographed, and the photos are the vouchers that are available in the Barcode of Life Data System (BOLD, http://boldsystems.org, accessed on 1 October 2023). The specimens were subsequently placed in a well containing 50 mL of 90% ethanol in a 96-well microplate and sent to the CCDB. Mites were sequenced for the barcode region of the COI gene according to standard protocols at the CCDB (www.ccdb.ca, accessed on 1 October 2023, using either LepF1/LepR1 [30] or LCO1490/m HCO2198 [31] primer pairs. The DNA extracts were placed in archival storages at −80 °C, most at the CCDB, and some (with sequencing code starting with UMNFO) at the University Museum of Bergen (ZMBN). Fifteen COI sequences that met the criteria of animal barcodes (sequence length ≥ 500 bp) were obtained. These sequences were blasted against GenBank in order to detect and exclude possible contaminations and were further used in the analyses. The sequences are available in GenBank (accessions numbers in Table 1).

Sequence variation within *N*. *coronata* specimens and between species was calculated in BOLD using the Kimura 2 Parameter distance model, pairwise deletion and BOLD Aligner (Amino Acid based HMM). The sequences were aligned by eye and neighbor-joining trees were constructed using MEGA6 [32]. Joint neighborhood topologies were visualized in FigTree 1.4.2 (available at http://tree.bio.ed.ac.uk/software/figtree).

### 2.3. Ecological Studies

Ecological studies on *N*. *coronata* were carried out in Atlantic raised bogs located in Hitra (coordinates were given above) and Høstadmyra (Trondheim municipality, Trøndelag, Norway, 63°24′19.4” N, 10°07′13.5” E, 110 m a. s. l.). Hitra is an island and forms a separate municipality, which is characterized by the mild oceanic climate, with a mean annual temperature of 8 °C and an annual precipitation of 917.8 mm. In the coldest month (January), the average temperature is 1 °C, and in the warmest month (July), it is 16 °C. Høstadmyra is characterized by a slightly colder and drier climate than in Hitra. The mean annual temperature is 6 °C, and annual precipitation is 575.6 mm. The average temperature in the coldest month (January) is −1 °C, and in the warmest month (July), it is 15 °C (https://www.timeanddate.no, accessed on 1 October 2023). In total, 63 samples of *Sphagnum* mosses of 500 cm^3^ each were collected on 29 and 30 June 2020 (26 from Hitra and 37 from Høstadmyra) from the following microhabitats: hummocks (31 samples; 12 from Hitra and 19 from Høstadmyra), lawns (14 samples; 4 from Hitra and 10 from Høstadmyra), low part of hummocks (4 samples; 2 from Hitra and 2 from Høstadmyra) and hollows (14 samples; 8 from Hitra and 6 from Høstadmyra). The method of extraction of samples was described above.

Populations of *N*. *coronata* from Hitra and Høstadmyra, and from studied microhabitats (hummocks, lawns, low part of hummocks and hollows) were characterized by abundance (*A* in 500 cm^3^). The basic statistical descriptors included the mean values and standard deviation. Equality of variance was tested with the Levene test, and normality of the distribution was tested with the Kolmogorov–Smirnov test. As the assumptions of variance analysis were not met, non-parametric tests were employed. Kruskal–Wallis ANOVA by ranks was utilized to test for significant differences between means [33]. The significance level for all analysis was accepted α = 0.05. These calculations were carried out with STATISTICA 12.5 software.

## 3. Results

### 3.1. Morphological Ontogeny of Nanhermannia coronata Berlese, 1913 (Figure 1, Figure 2, Figure 3, Figure 4, Figure 5, Figure 6, Figure 7, Figure 8, Figure 9, Figure 10, Figure 11, Figure 12, Figure 13, Figure 14, Figure 15, Figure 16, Figure 17, Figure 18, Figure 19, Figure 20, Figure 21, Figure 22 and Figure 23)

#### 3.1.1. Diagnosis

Adults are of medium size (length 450–660), with characters of *Nanhermannia* given by Seniczak et al. [2]. The prodorsal seta *in* is thin, longer than the bothridial seta; protuberances on the posterior part of the prodorsum are highly sclerotized, with 5–7 small posterior tubercles and five pairs of light spots between setal pair *in*. Seta *exp* is reduced to its alveolus. The ratio of body length/width is 2.3:1, and the notogastral setae are long, *c*_1_, *c*_3_, *d*_2_ and *e*_1_ reaching insertions of setae *d*_1_, *d*_2_, *e*_1_ and *h*_1_, respectively. The formula of the epimeral setae is 3-1-3-4. Seta *d* accompanying solenidion σ on all genua, φ_1_ on tibia I and φ on other tibiae are present.

Juveniles are elongated, the body unpigmented and with pits, the hysterosoma cylindrical and the central part of the prodorsum, epimeres and legs light brown. The bothridium is small, the bothridial seta absent, seta *exp* short and *exa* reduced to its alveolus. The larva has 12 pairs of short gastronotal setae, including *f*_1_ and *h*_2_; nymphs have 15 pair of long setae, excluding *f*_1_ reduced to its alveolus, all setae smooth. Leg segments are relatively thick and oval in cross section, and many setae on tarsi are relatively short, thick and conical, except for the longer apical setae. Seta *d* accompanying solenidion σ on all genua, φ_1_ on tibia I and φ on other tibiae are present.

The formula of the genital setae is 1-4-6-9 (protonymph to adult), and femora of deutonymph are 4-4-2-2 (leg I–IV), tritonymph 5-(5-6)-(2-3)-3 and adult 5-7-3-3.

**Figure 1 animals-13-03590-f001:**
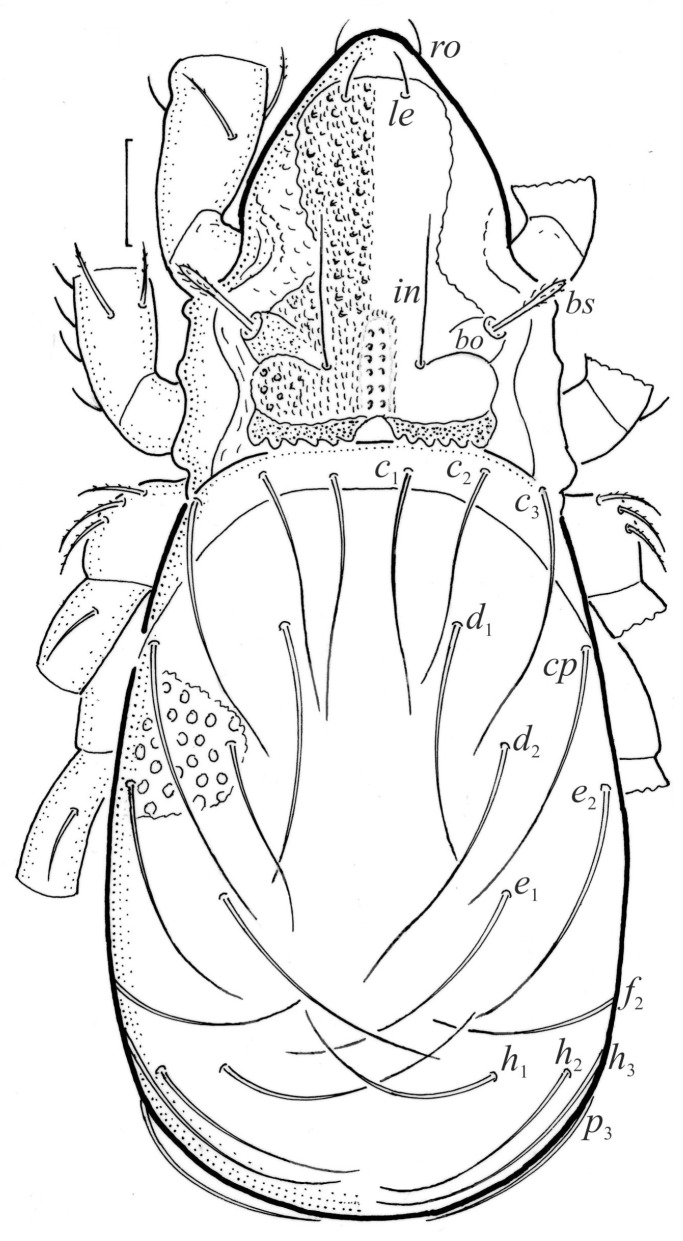
*Nanhermannia coronata*, adult, legs partially drawn, scale bars 50 μm. Dorsal aspect.

**Figure 2 animals-13-03590-f002:**
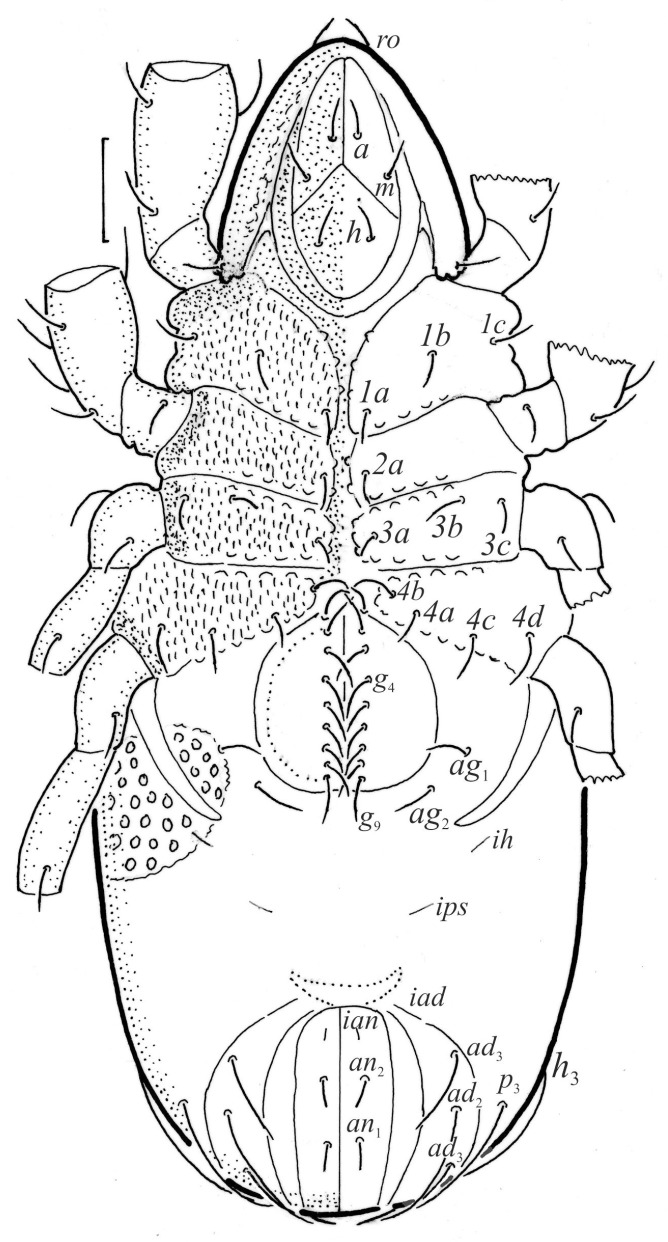
*Nanhermannia coronata*, adult, legs partially drawn, scale bars 50 μm. Ventral aspect.

**Figure 3 animals-13-03590-f003:**
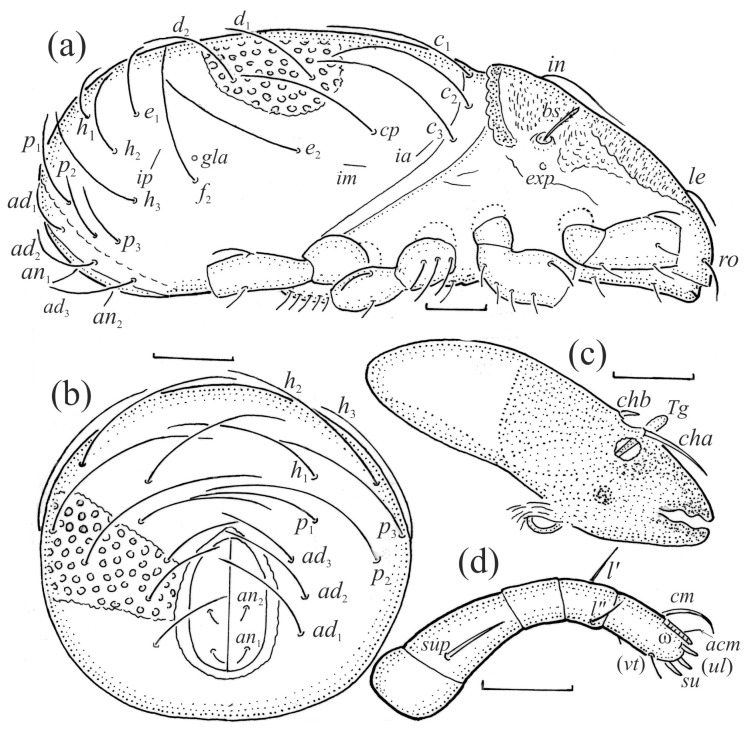
*Nanhermannia coronata*, adult. (**a**) Lateral aspect, legs partially drawn. (**b**) Posterior part of notogaster, posterior aspect. Mouthparts, right side, antiaxial aspect. (**c**) Chelicera. (**d**) Palp. Scale bars (**a**–**c**) 50 μm, (**d**) 20 μm.

**Figure 4 animals-13-03590-f004:**
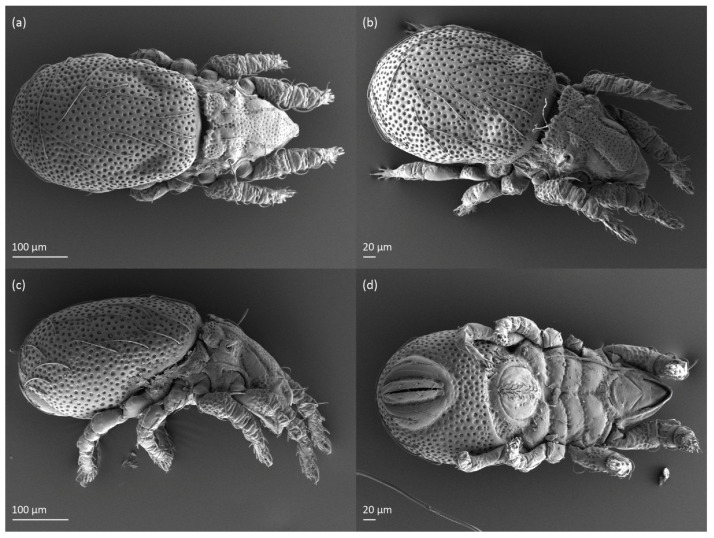
*Nanhermannia coronata,* adult, SEM micrographs. (**a**) Dorsal view, (**b**) dorsolateral view, (**c**) lateral view, (**d**) ventral view.

**Figure 5 animals-13-03590-f005:**
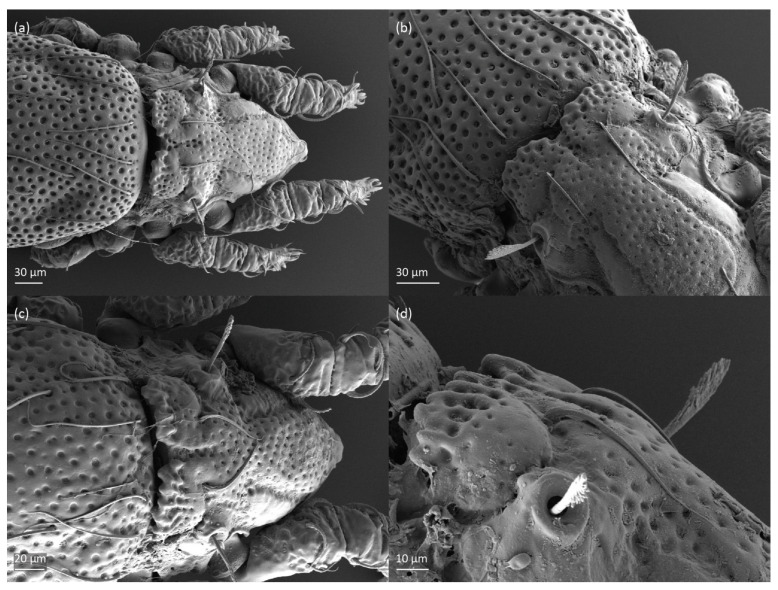
*Nanhermannia coronata,* adult, SEM micrographs. (**a**–**c**) Anterior part of body, dorsal view, (**d**) bothridial seta, lateral view.

**Figure 6 animals-13-03590-f006:**
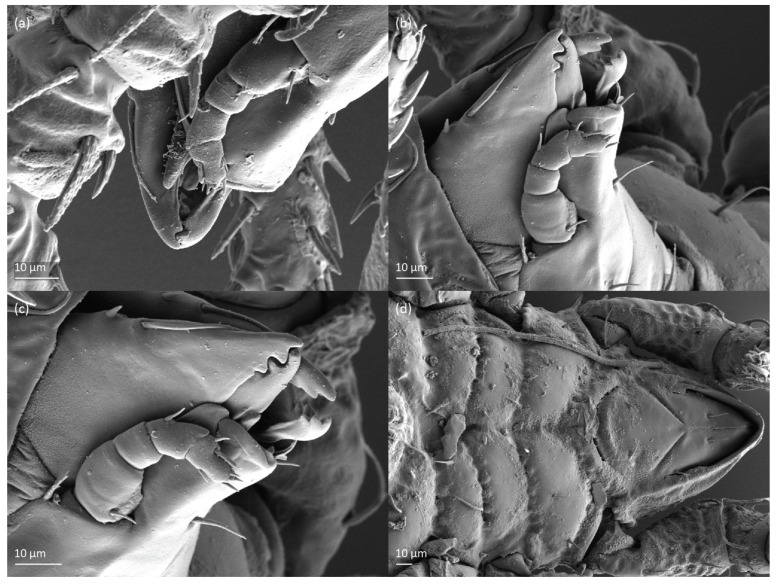
*Nanhermannia coronata,* adult, SEM micrographs. (**a**–**c**) Mouthparts, lateral view, (**d**) anterior part of body, ventral view.

**Figure 7 animals-13-03590-f007:**
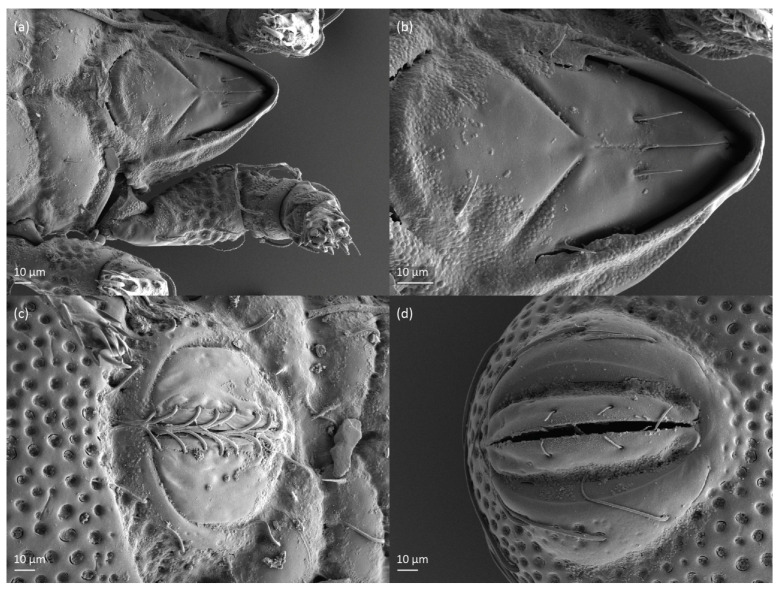
*Nanhermannia coronata,* adult, SEM micrographs. Ventral view, (**a**,**b**) anterior part of body, (**c**) genital plates, (**d**) anal plates.

**Figure 8 animals-13-03590-f008:**
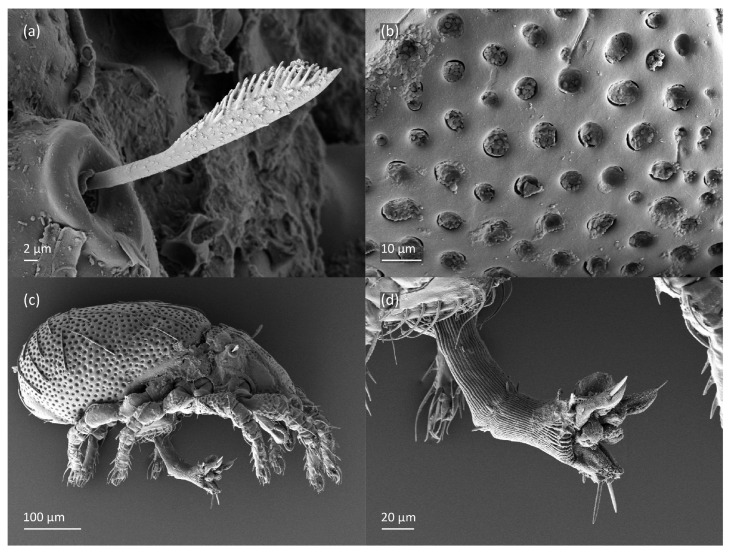
*Nanhermannia coronata*, adult, SEM micrographs. (**a**) Bothridial seta, lateral view; (**b**) pattern of notogaster; lateral view, (**c**) adult with rejected ovipositor, (**d**) ovipositor.

**Figure 9 animals-13-03590-f009:**
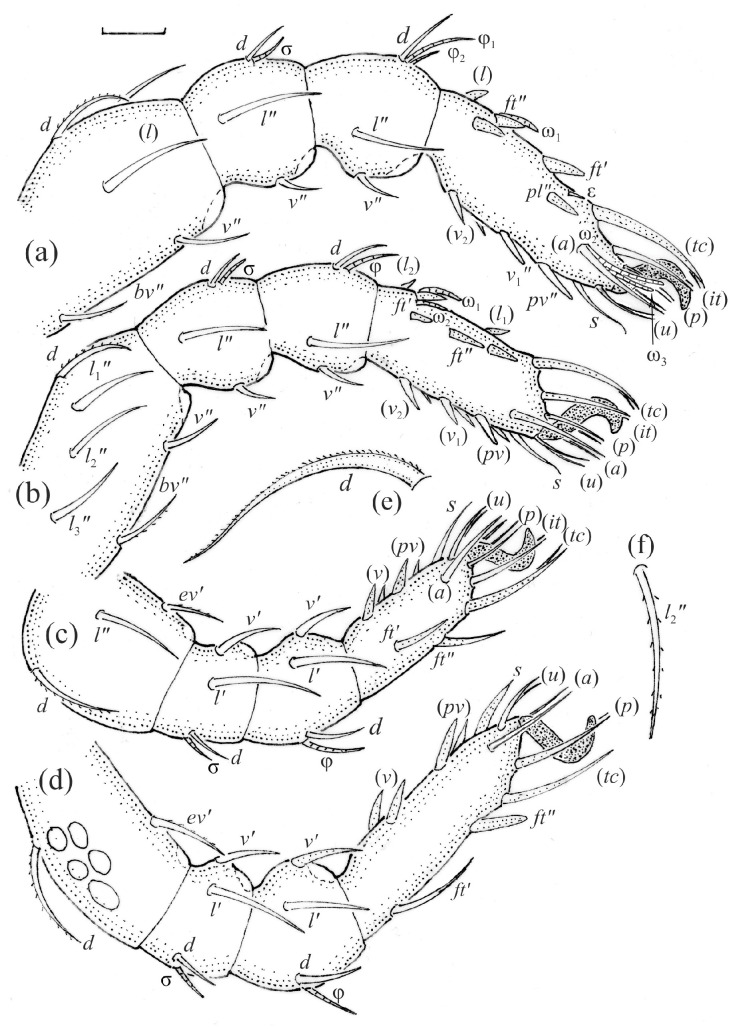
*Nanhermannia coronata*, leg segments of adult (part of femur to tarsus), right side, antiaxial aspect, setae on the opposite side not illustrated are indicated in the legend, scale bar 20 μm. (**a**) Leg I, genu (*l’*, *v’*), tibia (*l’*, *v’*), tarsus (*pl’*, *v*_1_*’*, *pv’*); (**b**) leg II, femur (*l*_1_*’*), genu (*l’*, *v’*), tibia (*l’*, *v’*); (**c**) leg III, tibia (*v’’*); (**d**) leg IV, tibia (*v’’*); (**e**) seta *d* on leg IV; (**f**) seta *l*_2_*’’* on femur II (e, f enlarged).

**Figure 10 animals-13-03590-f010:**
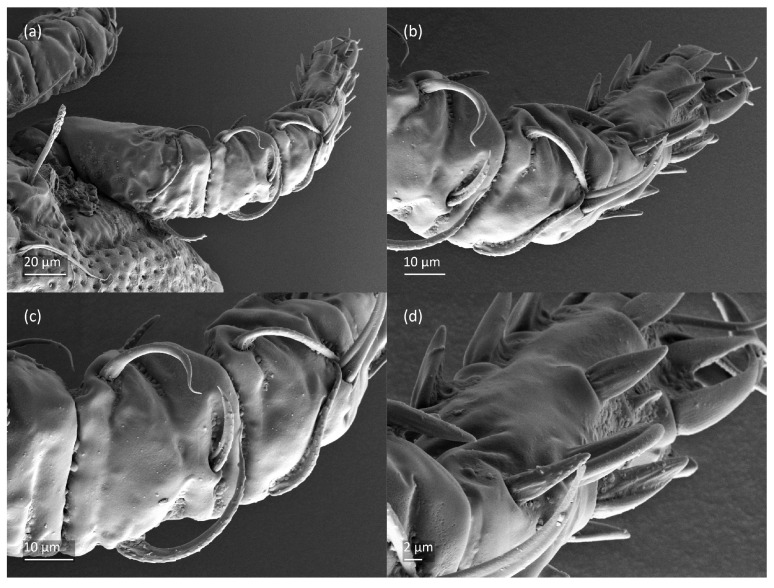
*Nanhermannia coronata*, adult, SEM micrographs. Dorsal view, (**a**) leg I; (**b**–**d**) parts of leg I.

**Figure 11 animals-13-03590-f011:**
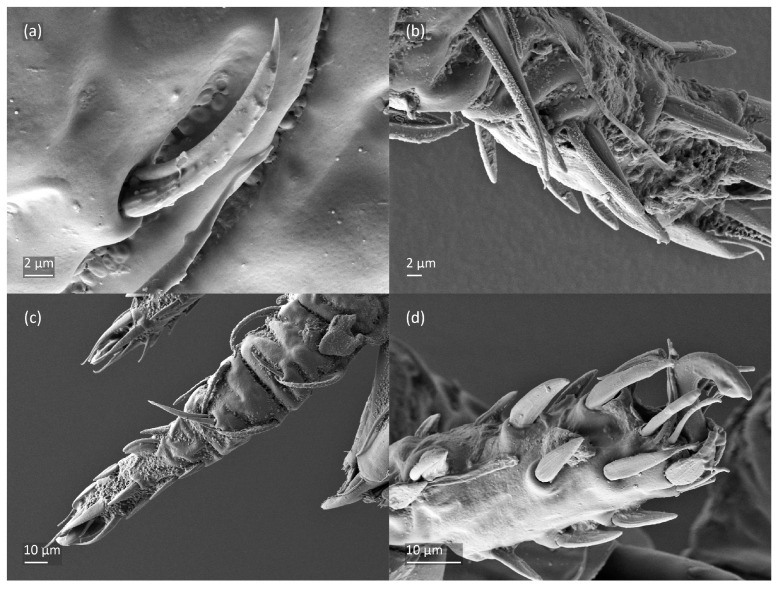
*Nanhermannia coronata*, leg I of adult, SEM micrographs. Dorsal view, (**a**) solenidion σ and seta *d* on genu I, (**b**) part of genu, tibia and tarsus I, (**c**) part of leg I; (**d**) part of tarsus I, lateral view.

**Figure 12 animals-13-03590-f012:**
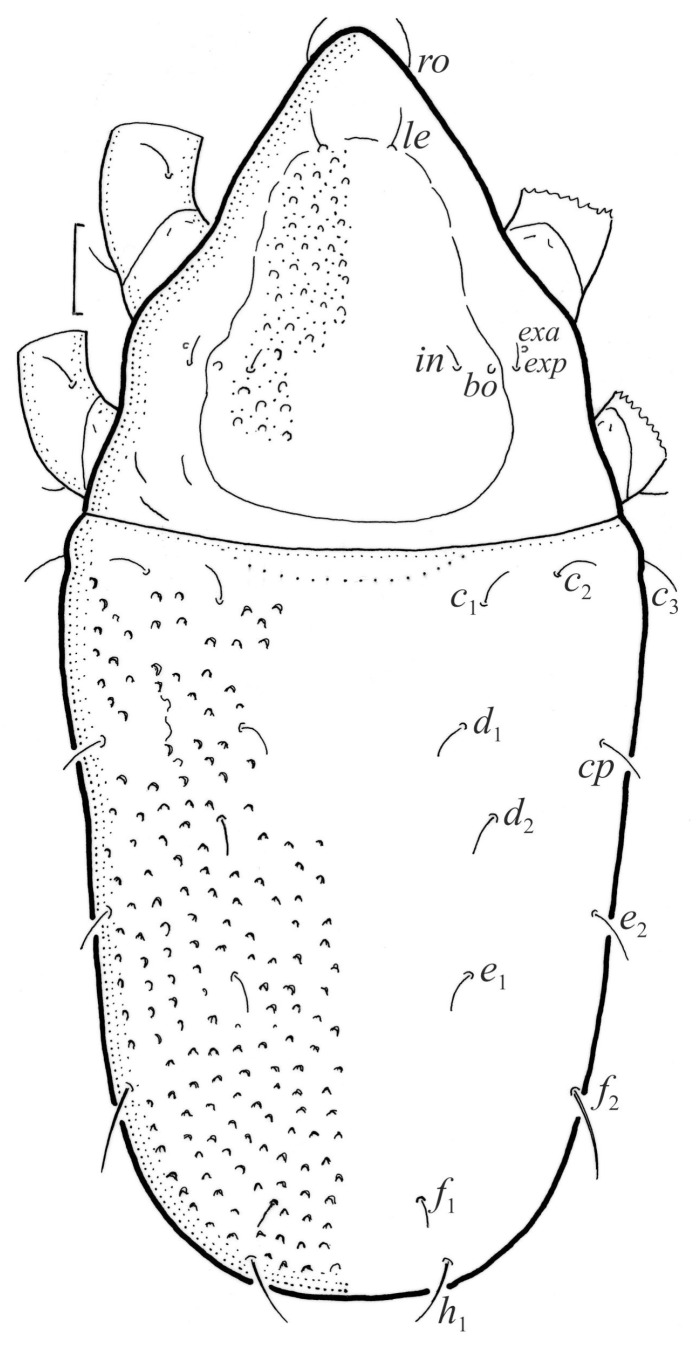
*Nanhermannia coronata*. Larva, dorsal aspect, legs I and II partially drawn, scale bar 20 μm.

**Figure 13 animals-13-03590-f013:**
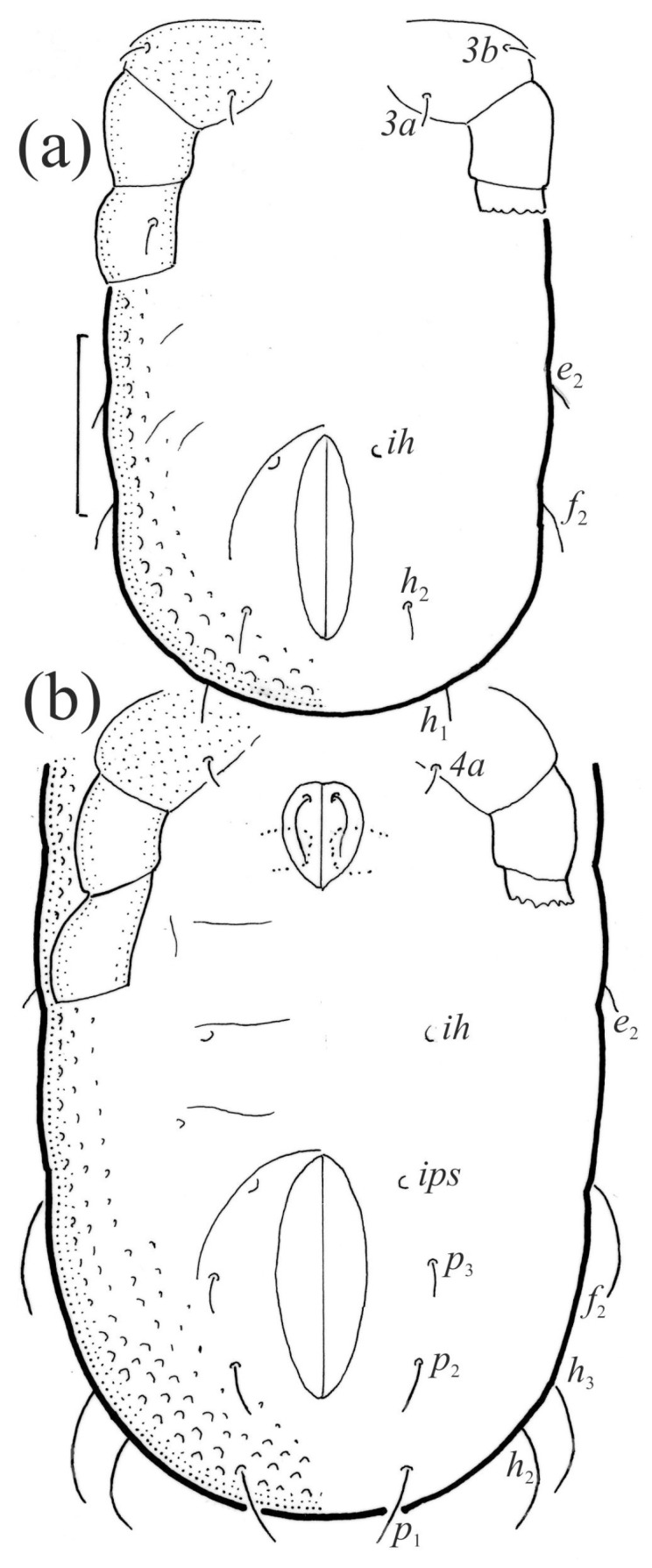
*Nanhermannia coronata*. Ventral part of hysterosoma, legs III and VI partially drawn, scale bar 50 μm, (**a**) larva, (**b**) protonymph.

**Figure 14 animals-13-03590-f014:**
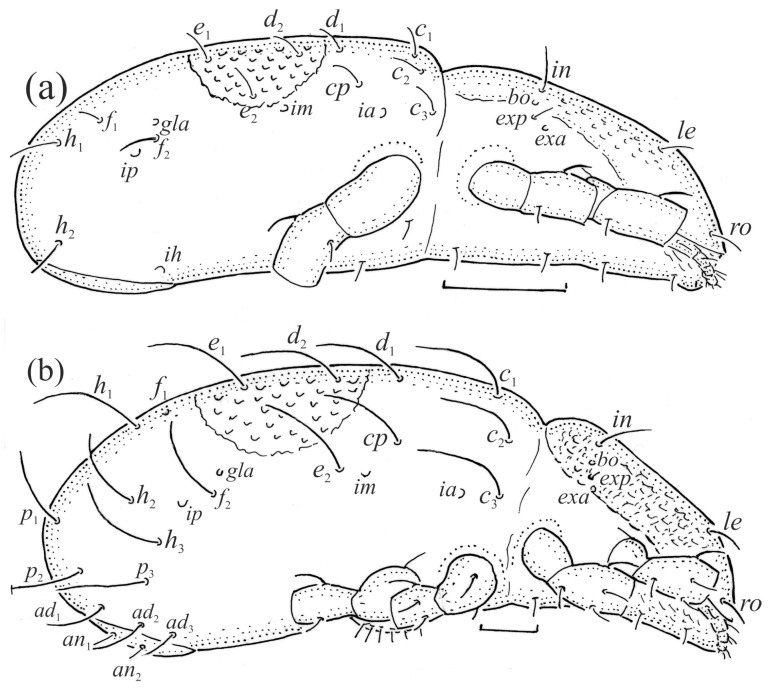
*Nanhermannia coronata*, lateral aspect, legs partially drawn, scale bars 50 μm. (**a**) Larva, (**b**) tritonymph.

**Figure 15 animals-13-03590-f015:**
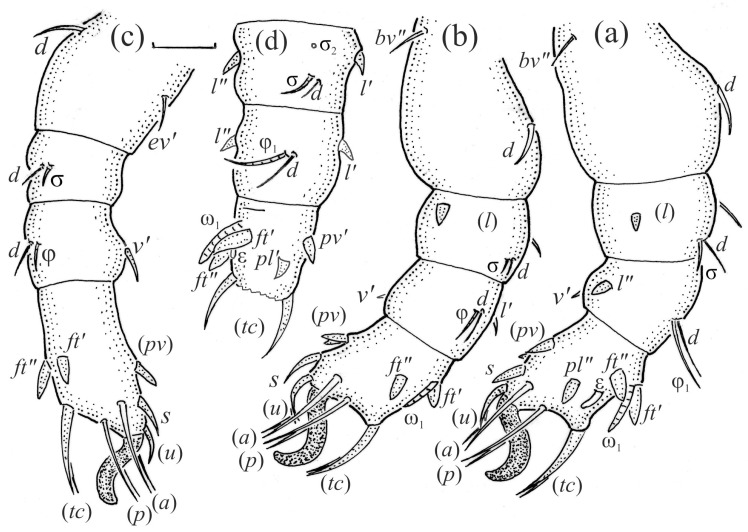
*Nanhermannia coronata*, leg segments of larva (part of femur to tarsus), right side, scale bar 20 μm. Antiaxial aspect, (**a**) leg I, tarsus (*pl’* not illustrated); (**b**) leg II; (**c**) leg III; dorsal aspect, (**d**) leg I, genu, tibia and part of tarsus.

**Figure 16 animals-13-03590-f016:**
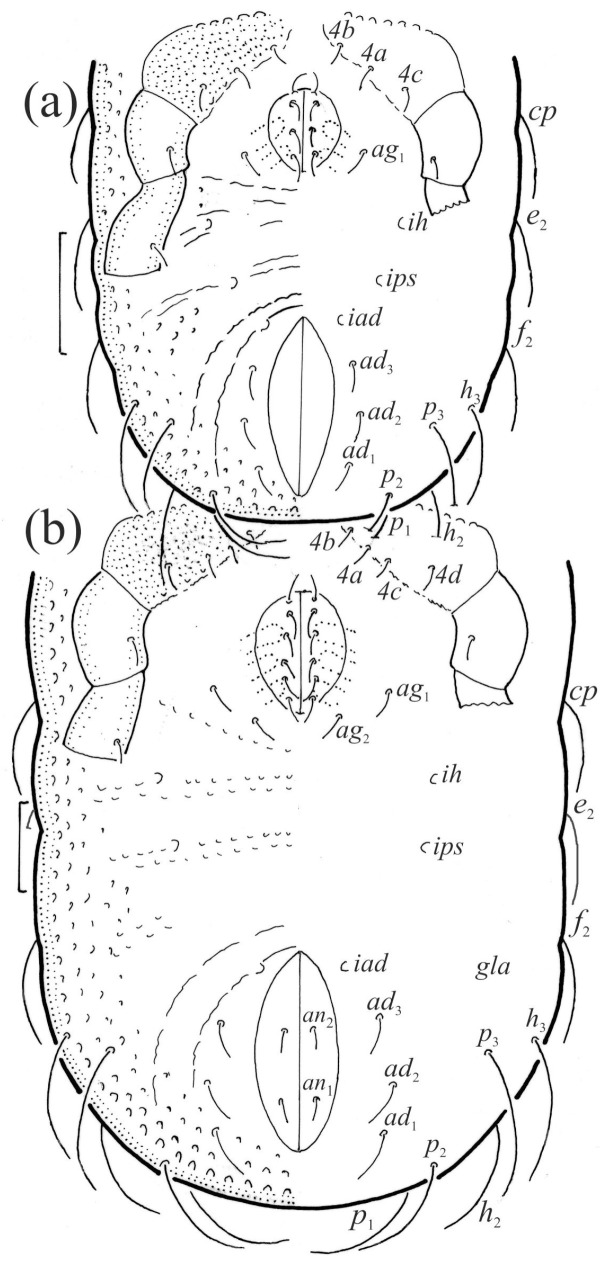
*Nanhermannia coronata*, scale bars 50 μm. Ventral part of hysterosoma, legs IV partially drawn, (**a**) deutonymph, (**b**) tritonymph.

**Figure 17 animals-13-03590-f017:**
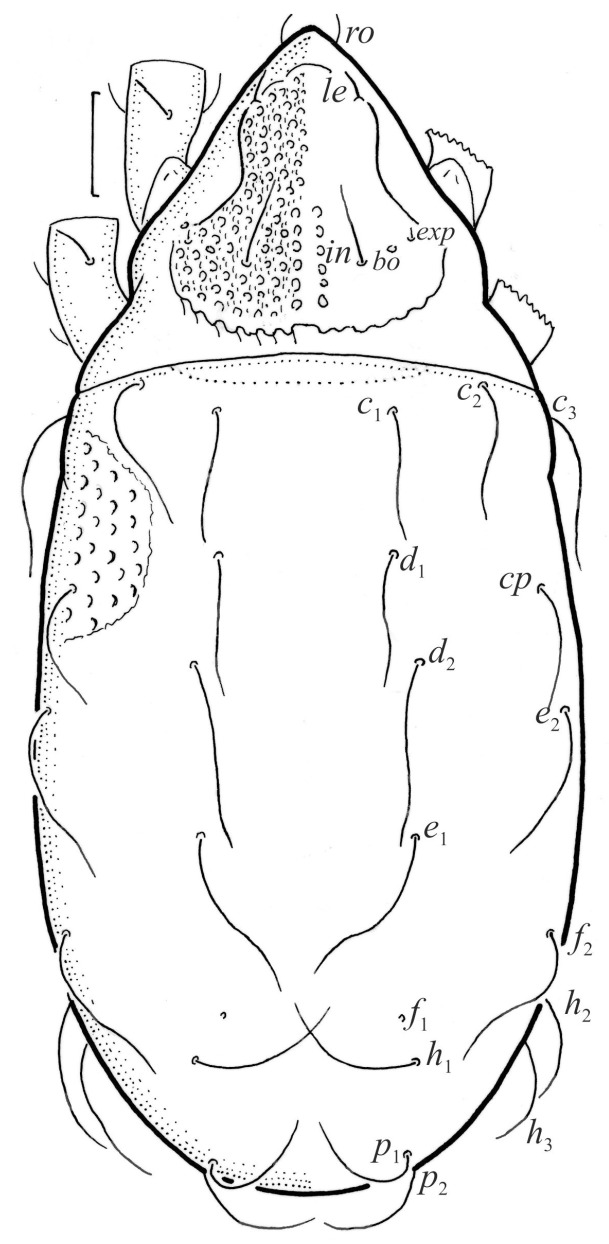
*Nanhermannia coronata*, scale bars 50 μm. Tritonymph, dorsal aspect, legs I and II partially drawn.

**Figure 18 animals-13-03590-f018:**
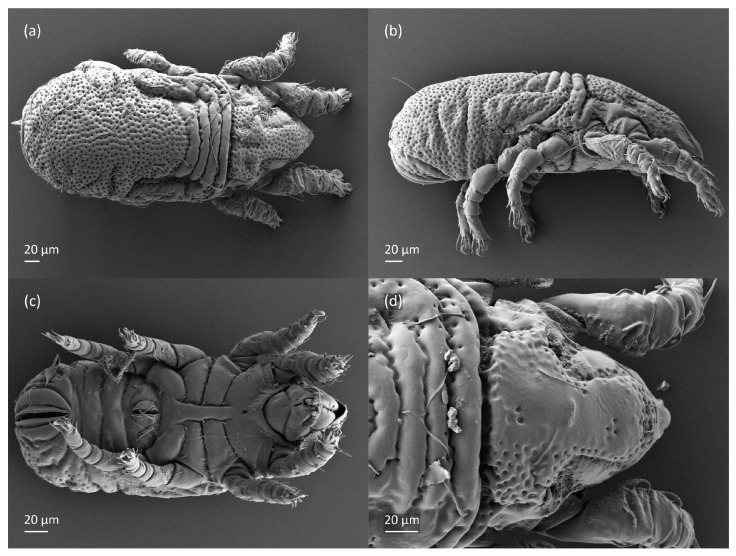
*Nanhermannia coronata*, tritonymph, SEM micrographs. (**a**) Dorsal view, (**b**) lateral view, (**c**), ventral view, (**d**) anterior and medial part of body.

**Figure 19 animals-13-03590-f019:**
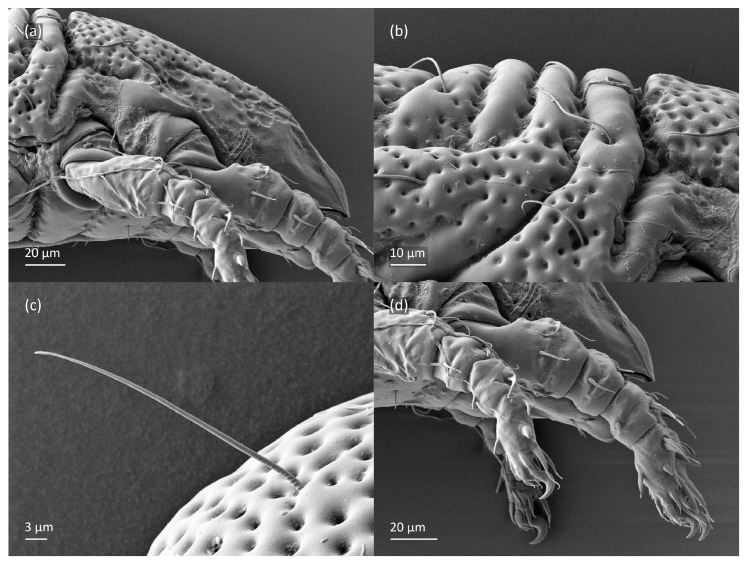
*Nanhermannia coronata*, tritonymph, SEM micrographs. Lateral view, (**a**) anterior part of body, (**b**) medial part of body, (**c**) seta *h*_3_, (**d**) leg I and II.

**Figure 20 animals-13-03590-f020:**
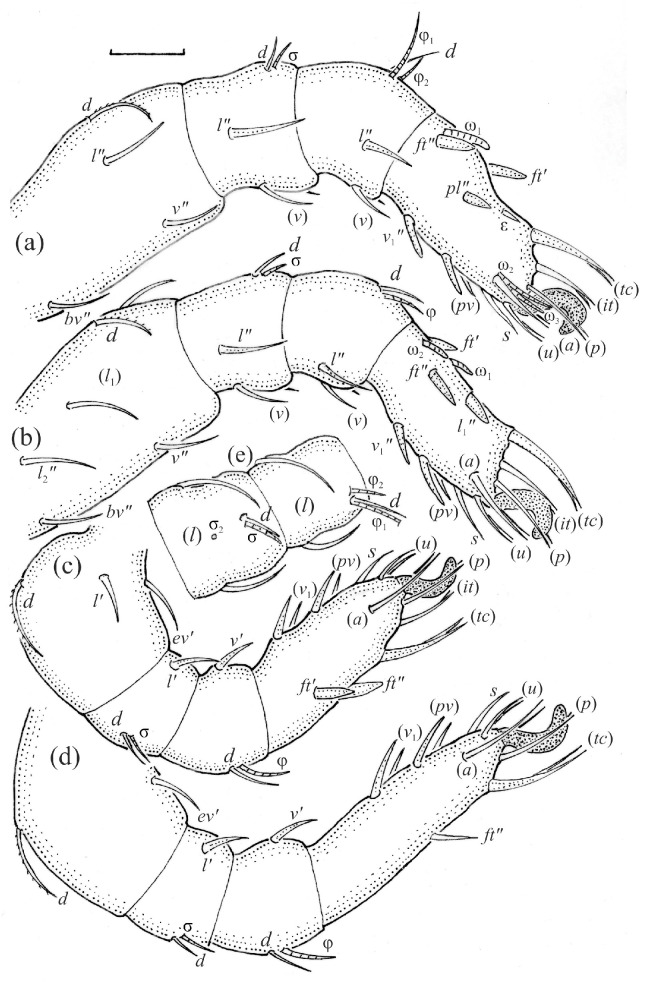
*Nanhermannia coronata*, leg segments of tritonymph (part of femur to tarsus), right side, seta on the opposite side not illustrated are indicated in the legend, scale bar 10 μm. Antiaxial aspect, (**a**) leg I, femur (*l’*), genu (*l’*), tibia (*l’’*), tarsus (*pl’*, *v’*); (**b**) leg II, genu (*l’*), tibia (*l’*), tarsus (*l*_1_*’*, *v*_1_*’*); (**c**) leg III, tibia (*v’’*); (**d**) leg IV, tibia (*v’’*); dorsal aspect, (**e**) genu and tibia I, dorsal view.

**Figure 21 animals-13-03590-f021:**
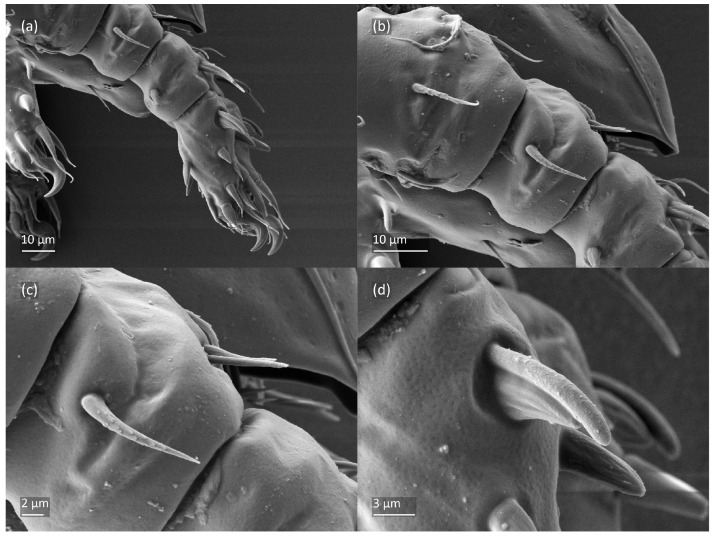
*Nanhermannia coronata*, tritonymph, SEM micrographs. Lateral view, (**a**–**d**) parts of leg I.

**Figure 22 animals-13-03590-f022:**
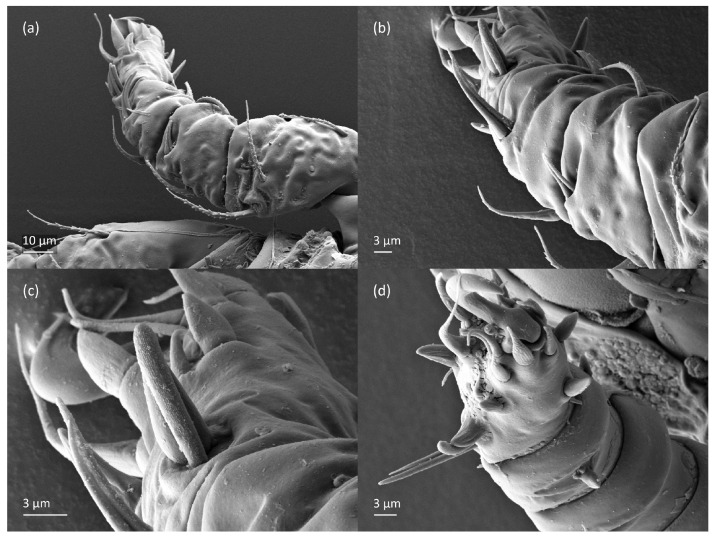
*Nanhermannia coronata*, tritonymph, SEM micrographs. Parts of leg I, (**a**–**c**) dorsal view; (**d**) ventral view.

**Figure 23 animals-13-03590-f023:**
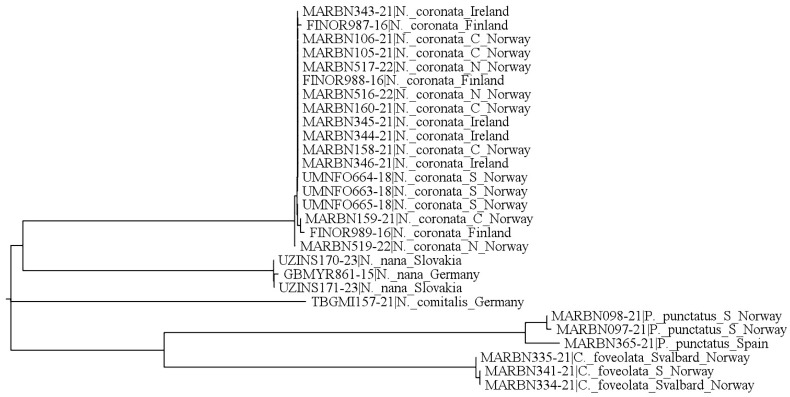
Neighbor-joining tree based on COI sequences of *Nanhermannia coronata* from southern, central and northern Norway (S Norway, C Norway and N Norway, respectively), Ireland, and Finland, and some other *Nanhermannia* species; *Platynothrus punctatus* and *Camisia foveolata* were used as outgroups. Details on sampling locations are given in Table 1.

#### 3.1.2. Morphology of Adult

The adult is elongated (Figure 1, Figure 2, Figure 3a and Figure 4), similar to that described by Berlese [34], but see Remarks. The mean length (and range) of females is 541.7 ± 14.0 (520–556, n = 30); the mean width (and range) is 252.0 ± 6.3 (228–260); males absent. The prodorsal seta *in* is long and thin, longer than the bothridial seta; protuberances on the posterior part of the prodorsum are highly sclerotized, with 5–7 small posterior tubercles (Figure 1, Figure 3a, Figure 4a–c and Figure 5 and Table 2), and six pairs of light spots present between setal pair *in*. The ratio of body length/width is 2.3:1. Seta *exp* short, *exa* is reduced to its alveolus. The prodorsum and hysterosoma are characterized by pits, observed in SEM figures as small holes (Figure 4, Figure 5, Figure 7c,d, Figure 8b,c and Figure 10a). The notogastral setae are long and smooth, *c*_1_, *c*_3_, *d*_2_ and *e*_1_ reaching insertions of setae *d*_1_, *d*_2_, *e*_1_ and *h*_1_, respectively (Figure 1, Figure 3a,b, Figure 4a–c, Figure 5a–c and Figure 8c and Table 2). Lyrifissure *ia* is posterior to seta *c*_3_, *im* anterior to seta *e*_2_, *ip* anterior to seta *h*_2_, *ian* anterior to seta *an*_2_, *iad* anteromedial to seta *ad*_3_, and *ips* and *ih* pushed anterior and anterolateral to seta *ad*_3_, respectively (Figure 2 and Figure 3a). Opisthonotal gland opening *gla* is not observed among pits. The chelicera is chelate-dentate, and seta *cha* is located anterior to *chb* and clearly longer than *chb*, both smooth (Figure 3c and Figure 6a–c). Most palpal setae are relatively short and smooth (Figure 3d and Figure 6a–c), solenidium ω and eupathidia are short, and the formula of setae (trochanter to tarsus + solenidion ω) is 1-0-2-7(1). Hypostomal and epimeral setae are short and smooth (Figure 2, Figure 4d, Figure 6c,d and Figure 7a,b). Aggenital setae (two pairs) are short, and genital setae (nine pairs) are slightly longer, all smooth (Figure 2, Figure 4d, Figure 7c,d and Figure 8c,d). Adanal setae (three pairs) are long, and anal setae are short, all smooth (Figure 2, Figure 3a,b, Figure 4d, Figure 7d and Figure 8c). The ovipositor is relatively thick, with relatively thick setae (Figure 8c,d). The legs are relatively thick, cuticle with ornamentation (Figure 4, Figure 5a–c, Figure 6c, Figure 7a, Figure 8c, Figure 9, Figure 10 and Figure 11) and all femora oval in cross section. Seta *l* on trochanter II and *d* on all femora are barbed, while other leg setae are smooth or finely barbed. Solenidion ω_1_ on tarsus I is located medial to seta *ft’’*, whereas solenidia ω_2_ and ω_3_ are located anterior to seta *a’’*. Solenidia ω_1_ and ω_2_ on tarsus II are located medial and lateral to seta *ft’*, respectively. Seta *d* accompanying solenidion σ on all genua, φ_1_ on tibia I and φ on other tibiae are present (Figure 9, Figure 10 and Figure 11 and Table 3). In all tarsi, hypertrichy occurs; setae on the basal and medial part of tarsi are conical, those on the distal part normal. The formulae of leg setae (and solenidia), trochanter to tarsus, are I–1-5-5(1)-5(2)-24(3), II–1-7-5(1)-5(1)-23(2), III–4-3-3(1)-4(1)-17 and IV–1-3-3(1)-4(1)-(14-15). Leg tarsi are monodactylous.

Remarks. The mean body length and width of individuals studied herein are larger than those described by Berlese [34]—length 490, width 220—but smaller than those investigated by Sitnikova [5]—length 575, width 250—and Weigmann [8]—length 480–570. Some authors [5,6,9,10] observed on protuberance of *N*. *coronata*, 4–5 posterior tubercles of different shape, whereas our adults have 5–7 small tubercles.

#### 3.1.3. Description of Juvenile Stages

The larva is elongated (Figure 12, Figure 13a and Figure 14a), the body unpigmented and with pits and the central part of the prodorsum, epimeres and legs light brown. The prodorsum is subtriangular, the central part punctate and with small pits. Prodorsal setae *ro*, *le*, *in* and *exp* are short and smooth (Figure 12 and Figure 14a and Table 2), and seta *exa* is reduced to its alveolus. The mutual distance between setal pair *le* is slightly shorter than that between setal pair *ro*, and the mutual distance between setal pair *in* is about two times longer than that between setal pair *ro*. The opening of the bothridium is small and rounded, and the bothridial seta is absent. The prodorsum and hysterosoma of the larva have small pits.

The hysterosoma of the larva is cylindrical, and the gastronotum has 12 pairs of setae, including dorsal *f*_1_ and ventral *h*_2_, inserted lateral to the posterior part of the anal valves (Figure 13a and Figure 14a); most are short and smooth, except for slightly longer *f*_2_ and *h*_1_ (Table 2). Cupule *ia* is posterior to seta *c*_3_, cupule *im* anterolateral to seta *e*_2_, cupule *ip* posterolateral to seta *f*_2_ and cupule *ih* lateral to the anterior part of the anal valves (Figure 13a and Figure 14a). The opisthosomal gland opening is medial to seta *f*_2_. The anal valves of the larva (segment P) are glabrous. The leg segments are relatively thick, and most leg setae are short, thick and conical, except for the longer apical setae on tarsi (Figure 15). Seta *d* accompanying solenidion σ on all genua, φ_1_ on tibia I and φ on other tibiae are present.

The shape and color of nymphs and prodorsal setae are as in the larva, but seta *in* is clearly longer, and pits in central part of prodorsum denser than in the larva. The bothridium is weakly developed and bothridial seta absent. The gastronotum of the protonymph has small pits and 15 pairs of setae because seta *f*_1_ has been lost and only the alveolus of this seta remains, and setae *h*_3_ and *p*-series appear and remain in the deutonymph and tritonymph (Figure 13b, Figure 14b, Figure 16, Figure 17 and Figure 18a–c). The prodorsum and hysterosoma of nymphs have pits, observed in SEM figures as small holes (Figure 18 and Figure 19). All gastronotal setae are long (Table 2) and smooth. In the protonymph, one pair of seta appears on the genital valves, and three pairs are added in the deutonymph and two pairs in the tritonymph (Figure 13b, Figure 14b and Figure 16), all short and smooth. In the deutonymph, one pair of aggenital setae appears, and one pair is added in the tritonymph, all short and smooth (Figure 14b and Figure 16a,b). Anal valves of the protonymph and deutonymph (segments AD and AN, respectively) are glabrous, and those of the tritonymph have two pairs of short and smooth setae. In the tritonymph, the opisthonotal gland opening and cupules *ia*, *im* and *ip* are as in the larva, cupule *iad* lateral to the anal valves, cupules *ips* and *ih* pushed anterolateral to cupule *iad* (Figure 14b and Figure 16b). The leg segments of the tritonymph relatively thick, and most leg setae are short, thick or conical, except longer apical setae on tarsi (Figure 18, Figure 19a,d, Figure 20, Figure 21 and Figure 22). Seta *d* accompanying solenidion σ on all genua, φ_1_ on tibia I and φ on other tibiae are present (Table 3).

#### 3.1.4. Summary of Ontogenetic Transformations

In the larva of *N*. *coronata*, the prodorsal setae *ro*, *le*, *in* and *exp* are short, and in the nymph, seta *in* is relatively longer, whereas in the adult, setae *ro* and *le* are short, *in* is long and *exp* is reduced to its alveolus. In all juveniles, the bothridium is weakly developed, and the bothridial seta is absent, whereas in the adult, the bothridium is well developed, with a small, rounded opening, and the bothridial seta is setiform, with a slightly thicker, barbed head. The larva has 12 pairs of gastronotal setae, including *f*_1_ and *h*_2_, whereas the nymphs and adult have 15 pairs (in the protonymph, *f*_1_ is reduced to its alveolus, and *h*_3_ and *p*-series are added). The formula of the gastronotal setae of *N*. *coronata* is 12-15-15-15-15 (larva to adult, excluding alveolar *f*_1_). The formulae of the epimeral setae are 3-1-2 (larva, including scaliform *1c*), 3-1-3-2 (protonymph), 3-1-3-3 (deutonymph) and 3-1-3-4 (tritonymph and adult). The formula of the genital setae is 1-4-6-9 (protonymph to adult), the aggenital setae is 1-2-2 (deutonymph to adult) and the formula of setae of segments PS–AN is 03333-0333-022. The ontogeny of leg setae and solenidia is given in Table 3.

### 3.2. Results of DNA Barcoding

A neighbor-joining tree based on cytochrome oxidase I (COI) nucleotide sequences confirmed morphological observations that the adults from all included localities (southern, central and northern Norway; Ireland; and Finland) represented the same species (Figure 23). The maximum mitochondrial DNA variation within *N. coronata* was 0.31%, while the minimum distance to compared representatives of putatively close genera was 26.53%.

### 3.3. Ecology and Biology

Our data on the ecology of *N. coronata* indicate that this species is common in raised bogs; it was present in 70% of collected samples. This study also shows wider ecological tolerance of *N. coronata* towards moisture; this species was found in different bog microhabitats: hummocks, lawns, transition zone between hummocks and hollows, and in hollows (Figure 24). It seems to prefer intermediate moisture conditions; in hummocks, it occurred in 87% of samples and was particularly abundant in the lower zone between hummocks and hollows (Figure 24). Results of Kruskal–Wallis ANOVA by ranks show significant differences between microhabitats (*H* = 9.27, *p* = 0.03).

In Hitra, the average abundance of *N. coronata* was higher (on average 4.42 specimens per 500 cm^3^) than in Høstadmyra (on average 2.73 specimens per 500 cm^3^) (Figure 25), but significant differences were observed only in deutonymphs (Kruskal–Wallis ANOVA by ranks: *H* = 3.96, *p* = 0.04). In Hitra, the average abundance of deutonymphs was 0.73 specimens per 500 cm^3^, while in Høstadmyra, it was 0.19 specimens per 500 cm^3^. In Hitra, the percentage of juveniles was higher than in Høstadmyra (39% and 32% of all individuals of species, respectively).

The stage structure of *N*. *coronata* we investigated in both populations of this species from Høstadmyra and Hitra. In total, 213 individuals of *N. coronata* were found, including 7 larvae, 24 protonymphs, 24 deutonymphs, 21 tritonymphs and 137 adults. Among 30 individuals investigated, all were females, and 25% were gravid, carrying one or rarely two large eggs (215 × 119), constituting about 40% of the total body length of females.

### 3.4. Comparison of Morphology of Nanhermannia coronata with Congeners

Seniczak et al. [2] compared the morphology of adults of *Nanhermannia* species, which differ from one another mainly in the shape of posterior prodorsal protuberances and tubercles (from prominent tubercles to small ones), and the length of some setae on the prodorsum and notogaster. Some species differ also from one another by the number of genital setae and formula of epimeral setae, but in most species, these characters are unknown. The shape of posterior prodorsal protuberance and the number of tubercles are considered diagnostic in *Nanhermannia*, but in some species, the number of tubercles varies [3,4,5,6,7,8,9,10], lowering their diagnostic values. The adult of *N*. *coronata* differs from that of *N*. *sellnicki* by its darker body color and larger number of posterior tubercles (6–7 tubercles) than that of *N*. *sellnicki* (3–4 tubercles), but some authors [5,8,9,10] observed on this protuberance 4–5 posterior tubercles, so the number of these tubercles in these species overlaps and has a small diagnostic value.

In Table 4, we compared selected morphological characters of adults, tritonymphs and larvae of *N*. *coronata*, *N*. *comitalis*, *N*. *nana* and *N*. *sellnicki*. The adult of *N*. *coronata* differs from that of *N*. *sellnicki* by the length of setae in and d_1_ and the formula of setae on femora ([2], Table 4). The juveniles of these species differ from one another by the length of some setae, the number of posterior tubercles and the formula of femora of the tritonymph. The juveniles of *N*. *coronata* differ from those of *N*. *sellnicki* by the formula of femora of tritonymph and the shape of insertion of prodorsal seta in and all gastronotal and adanal setae. In the latter species, these setae are in small individual depressions [2], whereas in *N*. *coronata*, these depressions are absent. The juveniles of *N*. *nana* have clearly longer gastronotal setae than other species, both in the larva and nymphs.

## 4. Discussion

*Nanhermannia coronata* is a common and abundant oribatid species in peatlands, but the adult of this species is often mistaken for *N*. *sellnicki*, which deteriorates our knowledge on the ecology of both species. According to our observations, the adult of *N*. *coronata* differs from that of *N*. *sellnicki* by having a darker body color [2], which is diagnostic only for darker adults of *N*. *coronata*. Freshly emergent adults of both species are light brown. In our studies, the adult of *N*. *coronata* differs from that of *N*. *sellnicki* by the length of seta *in* and the number of posterior tubercles on prodorsal protuberance (5–7 and 3–4 tubercles, respectively), but this character also has a small diagnostic value. Some authors [5,6,9,10] observed on prodorsal protuberance of *N*. *coronata* 4–5 posterior tubercles, which overlap with the diagnostic character of *N*. *sellnicki*. Diagnostic for these species is the formula of setae on femora, which clearly differentiates the adult and tritonymph of *N*. *coronata* studied herein from those instars of *N*. *sellnicki* ([2], Table 4). All juvenile stages of *N*. *coronata* differ from those of *N*. *sellnicki* by the shape of insertion of prodorsal seta *in* and all gastronotal and adanal setae. In the latter species, these setae are inserted in small individual depressions, whereas in *N*. *coronata*, these depressions are absent.

In the juveniles of *N*. *coronata* studied herein, some morphological differences were observed, comparing to those investigated by Ermilov and Łochyńska [9]. In the larva studied herein, seta *f*_2_ is slightly longer and seta *h*_2_ slightly shorter than in that studied by Ermilov and Łochyńska [9], whereas the nymphs have most gastronotal setae slightly longer than those investigated by these authors. The nymphs studied herein have clear posterior prodorsal protuberance, which was not observed by the mentioned authors [9]. These differences may reflect either the geographic variability of juveniles of *N*. *coronata* or different methods of preparation of figures.

In *N*. *coronata,* seta *f*_1_ is lost in the protonymph, as in *N*. *sellnicki* [2] and most genera of Crotonioidea, except for *Hermannia* Nicolet, 1855, *Phyllhermannia* Berlese, 1916, and *Nothrus* C.L. Koch, 1836, in which this seta is retained in all instars [36,37,38,39,40,41]. The larva of *N*. *coronata* has a pit on dorsal part of genu I, as that of *N*. *sellnicki* [2], which Grandjean [27] considered an alveolus of second solenidion σ. Two solenidia on genu I occur in some groups of lower Oribatida, for example in Lohmanniidae [42,43].

The leg segments, setae and solenidia of *N*. *coronata* are generally similar as in *N*. *sellnicki* except for the pattern of sculpture on femora, the shape of some setae and the number of setae on femur II and tarsi III and IV [2]. In both species, most leg setae are short, thick or conical, and most solenidia are blunt. In these species, an additional seta *l* occurs on femora II and III, and *l* and *v* on most tarsi, as in *Platynothrus coulsoni* A. and S. Seniczak, 2022, *P*. *punctatus* (L. Koch, 1879) and *P*. *troendelagicus* Seniczak et al., 2022 [21,44,45]. However, in *N*. *coronata* and *N*. *sellnicki*, setal pairs *l*_1_ and *l*_2_ on tarsus II are separated by solenidia ω_1_ and ω_2_ and setae *ft*, whereas in *Platynothrus* species, solenidia ω_1_ and ω_2_ and setae *ft* are located in anterior position, and all setae *l* are added posterior to them. In all species of *Nanhermannia* and *Platynothrus*, the number of setae on femora of the deutonymph, tritonymph and adult, and the number of setae on tarsi of the adult is species-specific.

The shape of the chelicera and palp of adults of *N*. *coronata* is similar to *N*. *sellnicki* and other *Nanhermannia* species discussed by Seniczak et al. [2]. The chelicera is chelate-dentate and has short and thick seta *chb* located posterior to longer seta *cha*, and the palpal setae are short and thick, and solenidion ω is separated from seta *acm*.

*Nanhermannia coronata* is known from the Holarctic region [46]. It is hygrophilous [47], with some range of tolerance towards moisture [48], but according to Rajski [23], with narrow tolerance towards pH and preferences of its low values. The acidity is considered the main factor for abundant occurrence of *N*. *coronata*, and this species has been mainly reported from raised bogs and swamps [15,17,18,19,20,22,49,50,51,52,53,54,55,56,57], less from other acidic forest soils and heaths [8], and it was absent from eutrophic mires [49,58]. Some studies show however that it has been abundant in beech forests [59], deciduous and birch forests [60], and in Scots pine forests [61]. In contrast, in broadleaf forests in Norway, it was few and rare [62,63,64]. In forests and open heath, it was strongly negatively affected by reduced soil moisture [65] and clearly associated with soil (not litter) [66]. It was also found in fruiting bodies of bracket fungi, and its abundance and frequency increased with the degree of decay of fungi [67]. Single individuals were also found in bird plumage and bird nests [68]. It was dominant in 14 Danish spring areas, which may reflect its tolerance to low levels of calcium [69]. In studied gradients in mires, its abundance was neither affected by the moisture nor by the pH, when it ranged between 3.68 and 5.75 [70].

*Nanhermannia coronata* is a secondary decomposer and fungivorous feeder (feeds partly on fungi and partly on litter) [71,72]. It reproduces parthenogenetically [46]. Ermilov and Łochyńska [9] cultured *N.* cf. *coronata* in laboratory conditions (22–23 °C, 100% air humidity) and fed it with algae (*Protococcus* sp.) and raw potatoes. The adults fed on both types of food, whereas the juveniles preferred algae. The total development of this species lasted 105–124 days, and each stage developed during the following days (+ immovable stage between instars): egg (6–8), larva [(13–20) + (4–8)], protonymph [(11–31) + (5–11)], deutonymph [(12–26) + (6–12)] and tritonymph [(9–26) + (10–13)] to obtain the adult. The maturation time of *N*. cf. *coronata* lasts 21 weeks at 20 °C and 16 weeks at 22.5 °C. Its tolerance to temperature was tested in another experiment, and it was similar in adults and juvenile specimens: 38 °C for 4 h and 36 °C for 12 h [73].

Our ecological observations confirmed a common occurrence of *N. coronata* in raised bogs, with a high percentage of juvenile stages in populations. In two studied locations, Hitra and Høstadmyra, we observed significant differences in the abundance of deutonymphs and proportion of the juvenile stages was higher in Hitra than in Høstadmyra, which can be explained by the warmer (average annual temperature higher by 2 °C) and milder, oceanic climate in Hitra compared to Høstadmyra. More advanced development of species and higher proportion of juveniles has also been observed in another oribatid species, *Ceratozetes parvulus* Sellnick, 1922, collected from the same bogs [74]. In wetter microhabitats, *N. coronata* was not abundant and occurred with a low constancy. This is consistent with the observations from inundated bog habitats—edges of water bodies—where *N. coronata* was absent or very few in number [75]. In degraded, drier bogs, it was more abundant [76,77], but in completely dry bogs, it was absent [78]. When the bog was dried, the abundance of *N. coronata* decreased in the hummocks but, at the same time, increased in the hollows [79].

*Nanhermannia coronata* prefers acid and humid microhabitats [23], whereas *N*. *sellnicki* is less common than *N*. *coronata* and occurs in drier habitats [13], like birch forests, especially with understory formed by *Vaccinium* and *Empetrum*. In investigated habitats, like hummocks, lawns, transition zone between hummocks and hollows, and in hollows, *N. coronata* was particularly abundant in between hummocks and hollows, with abundant juvenile stages.

According to the literature, this species can also be abundant in beech forests [59], deciduous and birch forests [60], Scots pine forests [61], bird plumage and bird nests [68], but these data need confirmation using the diagnostic characters of species given herein.

## 5. Conclusions

1. *Nanhermannia coronata* is an abundant and common oribatid mite in raised bogs, with a high percentage of juveniles, and has a preference for humid microhabitats, whereas *N*. *sellnicki* is less common than *N*. *coronata* and occurs in drier habitats.

2. *Nanhermannia coronata* differs clearly from *N*. *sellnicki* by the following morphological characters: the number of setae on femora I–IV of the adult (*N*. *coronata* 5-7-3-3, *N*. *sellnicki* 5-6-3-2) and tritonymph [*N*. *coronata* 5-(5-6)-(2-3)-3, *N*. *sellnicki* 5-6-3-2], and the shape of insertions of prodorsal seta *in* and all gastronotal and adanal setae of juveniles; in *N*. *sellnicki*, these setae are inserted in small individual depressions, whereas in *N. coronata*, these depressions are absent.

## Figures and Tables

**Figure 24 animals-13-03590-f024:**
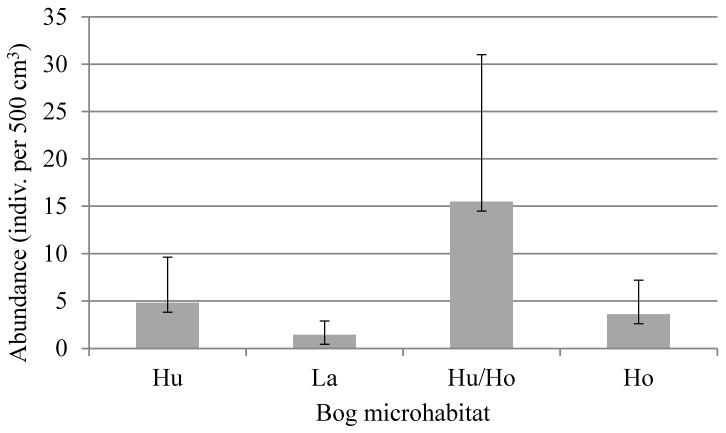
Average abundance of *Nanhermannia coronata* (individuals per 500 cm^3^) (bars) with standard deviation (whiskers) in selected microhabitats of bogs in Norway: Hu—hummock, La—lawn, Hu/Ho—transition zone between hummock and hollow, Ho—hollow; numbers above bars present number of samples collected from a certain microhabitat and percentage of samples where the species was present (constancy index).

**Figure 25 animals-13-03590-f025:**
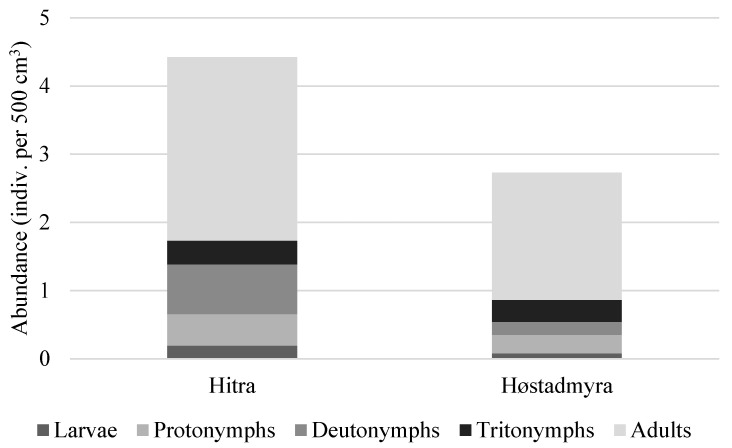
Average abundance of developmental stages of *Nanhermannia coronata* in two bogs in Norway.

**Table 1 animals-13-03590-t001:** Information about sequenced specimens of *Nanhermannia coronata* and other oribatid species used in this study; ad—adult, juv—juveniles.

Species	Sequence Code at BOLD	Stage	GeneBank Access No.	Locality	Coordinates	Elevation (m a.s.l.)	Collection Data
*N. coronata*	UMNFO663-18	ad	OR732229	NO: Vestland, Lydehorn	60.370, 5.244	216.8	6 October 2018, Seniczak, A.
	UMNFO664-18	ad	OR732235	NO: Vestland, Lydehorn	60.370, 5.244	216.8	6 October 2018, Seniczak, A.
	UMNFO665-18	ad	OR732231	NO: Vestland, Lydehorn	60.370, 5.244	216.8	6 October 2018, Seniczak, A.
	MARBN105-21	ad	OR732225	NO: Trondelag, Hitra	63.489, 8.874	48.2	29 July 2020, leg. A. Seniczak, K.I. Flatberg, K. Hassel, S. Roth
	MARBN106-21	ad	OR732221	NO: Trondelag, Hitra	63.489, 8.874	48.2	29 July 2020, leg. A. Seniczak, K.I. Flatberg, K. Hassel, S. Roth
	MARBN158-21	ad	OR732223	NO: Trondelag, Høstadmyra	63.405, 10.12	110.0	30 July 2020, leg. A. Seniczak, K.I. Flatberg, K. Hassel, S. Roth
	MARBN159-21	ad	OR732230	NO: Trondelag, Høstadmyra	63.405, 10.12	110.0	30 July 2020, leg. A. Seniczak, K.I. Flatberg, K. Hassel, S. Roth
	MARBN160-21	ad	OR732234	NO: Trondelag, Høstadmyra	63.405, 10.12	110.0	30 July 2020, leg. A. Seniczak, K.I. Flatberg, K. Hassel, S. Roth
	MARBN516-22	ad	OR732224	NO: Nordland, Kummeren	67.043, 14.216	28.1	26 July 2021, Seniczak, A., Flatberg, K.I.
	MARBN517-22	ad	OR732218	NO: Nordland, Kummeren	67.043, 14.216	28.1	26 July 2021, Seniczak, A., Flatberg, K.I.
	MARBN519-22	ad	OR732227	NO: Nordland, Kummeren	67.043, 14.216	28.1	26 July 2021, Seniczak, A., Flatberg, K.I.
	MARBN343-21	ad	OR732226	IR: Leinster, Lullymore	53.270, −6.949	77.4	9 December 2014, leg. A. Seniczak, T. Bolger
	MARBN344-21	ad	OR732233	IR: Leinster, Lullymore	53.270, −6.949	77.4	9 December 2014, leg. A. Seniczak, T. Bolger
	MARBN345-21	ad	OR732232	IR: Leinster, Lullymore	53.270, −6.949	77.4	9 December 2014, leg. A. Seniczak, T. Bolger
	MARBN346-21	ad	OR732228	IR: Leinster, Lullymore	53.270, −6.949	77.4	9 December 2014, leg. A. Seniczak, T. Bolger
	FINOR987-16	ad	MZ608481	FIN: Varsinais-Suomi: Paimio, Jaervessuo	60.451, 22.775	49.9	9 October 2014, leg. R. Penttinen
	FINOR988-16	ad	MZ609187	FIN: Varsinais-Suomi: Paimio, Jaervessuo	60.451, 22.775	49.9	9 October 2014, leg. R. Penttinen
	FINOR989-16	ad	MZ611116	FIN: Varsinais-Suomi: Paimio, Jaervessuo	60.451, 22.775	49.9	9 October 2014, leg. R. Penttinen
*N. nana*	GBMYR861-15	ad	In BOLD	GE:			
	UZINS170-23	ad	In BOLD	SL: Bratislava, Sitina	48.171, 17.0656	213.3	1 June 2022, leg. Mangova, B.
	UZINS171-23	ad	In BOLD	SL: Bratislava, Sitina	48.171, 17.0656	213.3	1 June 2022, leg. Mangova, B.
*Camisia foveolata*	MARBN334-21	ad	OR732222	NO: Svalbard, Longyearbyen, Endalen	78.209, 15.711	22.6	5 June 2018, Roth, S.
	MARBN335-21	juv	OR732220	NO: Svalbard, Longyearbyen, Endalen	78.209, 15.711	22.6	5 June 2018, Roth, S.
	MARBN341-21	ad	OR773185	NO: Vestland, Finse	60.593, 7.432	1352.0	7 September 2018, Seniczak, A.
*Platynothrus punctatus*	MARBN097-21	ad	OL671034	NO: Vestland, Finse	60.593, 7.432	1352.0	22 September 2019, Seniczak, A.
	MARBN098-21	ad	OL671021	NO: Vestland, Finse	60.593, 7.432	1352.0	22 September 2019, Seniczak, A.
	MARBN365-21	juv	OL671024	SP: Andalusia, Borreguil de la Virgen	37.087, −3.374	2500.7	18 August 2017, Seniczak, A., F. Ondoño, E.

**Table 2 animals-13-03590-t002:** Measurements of some morphological characters of juvenile stages and adult of *Nanhermannia coronata* (mean measurements of 10 specimens in μm); nd—not developed.

Morphological Characters	Larva	Protonymph	Deutonymph	Tritonymph	Adult
Body length	284	351	416	572	559
Body width	125	155	172	305	251
Length of prodorsum	104	120	152	171	215
Length of:					
seta *ro*	16	17	21	32	39
seta *le*	15	17	19	30	35
seta *in*	19	23	30	45	75
seta *bs*	nd	nd	nd	nd	67
seta *c*_1_	12	19	30	57	129
seta *c*_3_	14	22	31	82	136
seta *cp*	10	20	32	95	142
seta *d*_1_	12	21	33	70	125
seta *d*_2_	13	22	32	93	135
seta *e*_1_	10	20	35	88	130
seta *e*_2_	11	21	38	83	134
seta *f*_1_	8	lost	lost	lost	lost
seta *f*_2_	24	32	43	96	133
seta *h*_1_	20	22	30	80	120
seta *h*_3_	nd	23	31	95	133
seta *p*_1_	nd	8	19	72	122
seta *p*_3_	nd	16	30	83	130
genital opening	nd	26	33	46	83
anal opening	52	65	77	123	114

**Table 3 animals-13-03590-t003:** Ontogeny of leg setae (Roman letters) and solenidia (Greek letters) in *Nanhermannia coronata*.

Leg	Trochanter	Femur	Genu	Tibia	Tarsus
Leg I					
Larva	–	*d*, *bv’’*	*d*, (*l*), σ	(*l*), *v’, d*, φ_1_	(*ft*), (*tc*), (*pl*), (*p*), (*u*), (*a*), *s*, (*pv*), ε, ω_1_
Protonymph	–	–	–	–	ω_2_
Deutonymph	*v’*	(*l*)	–	φ_2_	–
Tritonymph	–	*v”*	*v’*, *v”*	*v’’*	(*v*_1_), (*it*), ω_3_
Adult	–	–	–	–	(*l*), (*v*_2_)
Leg II					
Larva	–	*d*, *bv’’*	*d*, (*l*), σ	*l’, v’, d*, φ	(*ft*), (*tc*), (*p*), (*u*), (*a*), *s*, (*pv*), ω_1_
Protonymph	–		–	*l’’*	–
Deutonymph	*v’*	(*l*_1_)	–	–	ω_2_
Tritonymph	–	*l*_2_*”*, *v”* ^1^	*v’*, *v”*	*v’’*	*l*_1′_, (*v*_1_), (*it*)
Adult	–	*l* _3′_	–	–	*l*_1_*”*, (*l*_2_), (*v*_2_)
Leg III					
Larva	–	*d*, *ev’*	*d*, σ	*v’*, *d*, φ	(*ft*), (*tc*), (*p*), (*u*), (*a*), *s*, (*pv*)
Protonymph	*v’*	–	–	–	–
Deutonymph	*l’* _1_	–	–	–	–
Tritonymph	*l’* _2_	*l’* ^1^	*v’*	*v’’*	(*it*), (*v*_1_)
Adult	*l’* _3_	–	–	*l’*	–
Leg IV					
Protonymph		–	–	–	*ft’’*, (*pv*), (*p*), (*u*)
Deutonymph	–	*d, ev’*	*l’*, *d*, σ	*v’*, *d*, φ	(*tc*), (*a*), *s*
Tritonymph	*v’*	*l’*	*v’*	*v”*	(*v*_1_)
Adult	–	–	–	*l’*	*ft’* ^2^

Note: structures are indicated where they are first added and are present through the rest of ontogeny; pairs of setae are in parentheses; dash indicates no additions; ^1^ is added in some individuals; if not, it is added in the next stage; ^2^ in some individuals is absent.

**Table 4 animals-13-03590-t004:** Comparison of selected morphological characters of some instars of *Nanhermannia coronata*, *N. comitalis*, *N. nana* and *N. sellnicki*.

Characters	*N*. *coronata*	*N*. *comitalis* ^1^	*N*. *nana* ^1^	*N*. *sellnicki* ^2^
Adult				
Length of seta *in*	Longer than *bs*	Longer than *bs*	As long as *bs*	As long as *bs*
Posterior tubercles ^3^	5–7	4	1	3–4
Length of seta *d*_1_ ^4^	No	Yes	Yes	No
Formula of femora	5-7-3-3	unknown	unknown	5-6-3-2
Tritonymph				
Length of seta *in*	As long as *c*_1_	As long as *c*_1_	Shorter than *c*_1_	As long as *c*_1_
Posterior tubercles ^3^	6–7	5	6	4–5
Length of seta *c*_1_ ^5^	No	Yes	Yes	No
Length of seta *d*_1_ ^4^	No	Yes	Yes	No
Length of seta *e*_1_ ^6^	No	No	Yes	No
Formula of femora	5-(5-6)-(2-3)-3	unknown	unknown	5-6-3-2
Larva				
Length of seta *in*	Shorter than *c*_1_	Longer than *c*_1_	Shorter than *c*_1_	As long as *c*_1_
Posterior tubercles ^3^	Absent	2–3	2	Absent
Length of seta *c*_1_ ^5^	No	Yes	Yes	No
Length of seta *d*_1_ ^4^	No	Yes	Yes	No

^1^ According to Seniczak [35], ^2^ Seniczak et al. [2], ^3^ on each posterior prodorsal protuberance, ^4^ reaches insertion of seta *e*_1_, ^5^ reaches insertion of seta *d*_1_, ^6^ reaches insertion of seta *h*_1_.

## Data Availability

Ecological data on *N. coronata* are stored at DataverseNO open archive (https://doi.org/10.18710/F7LO2Y).

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
