# Peer review of "Morphological Ontogeny and Ecology of a Common Peatland Mite, Nanhermannia coronata (Acari, Oribatida, Nanhermanniidae)"

_animals, 2023, doi:10.3390/ani13223590_

Round 1

Reviewer 1 Report

Comments and Suggestions for Authors

I have no comment for the authors.

Reviewer 2 Report

Comments and Suggestions for Authors

Manuscript ID: animals-2661148

Title: Morphological ontogeny and ecology of a common peatland mite, Nanhermannia coronata (Acari, Oribatida, Nanhermanniidae)

Authors: Stanisław Seniczak, Anna Beata Seniczak

The authors present a study on the morphological ontogeny and ecology of Nanhermannia coronata Berlese, 1913. It is a common and abundant oribatid species in peatlands, but can be easily mistaken with N. sellnicki Forsslund, 1958 as adult. Therefore, the identity of adults of N. coronata from several sites in Norway and Ireland was supported by the COI sequence data, and based on this material the morphological ontogeny of this species is described and illustrated to point the differences between N. coronata and N. sellnicki. The authors described from the morphological point of view all juvenile stages of N. coronate.  The authors found some morphological characters that differ clearly N. coronata from N. sellnicki, like the number of setae on femora of adult and tritonymph, and the shape of insertions of prodorsal seta in, and all gastronotal and ada-nal setae of juveniles; in N. sellnicki these setae are inserted in small individual depressions, whereas in N. coronata these depressions are absent. Their ecological observations confirm a common occurrence of N. coronata in raised bogs, high percentage of juvenile stages in its populations and preference of this species to humid microhabitats, whereas N. sellnicki is less common than N. coronata and occurs in drier habitats.

The information presented in the manuscript is new, original and well documented.

           The title of the manuscript reflects the content of the study.

           The abstract reflect the manuscript content.

           The manuscript is sustained by a suitable and complete literature.

           The information from the manuscript is well structured. The manuscript is complex. The description of Nanhermannia coronata is made taking into account the morphological, ecological and molecular data. All immature stages were morphological described!

           The introduction offers the proper arguments for the objectives of the study.  It is well scientifically argued.

           The manuscript aims are clear presented.

           The methodology chapter is detailed, proper and complete described.

           The discussion chapter is sustained by the results.

           The tables and figures reveal properly the described results/data.

           Some little observations were included in the manuscript text!

One comment: Please, insert more details about the distribution of Nanhermannia coronate in Europe and the world!

The manuscript is proper for Animals journal. 

Reviewer 3 Report

Comments and Suggestions for Authors

Dear Authors,

Present study deals with the ontogeny and ecological preferences of a common oribatid mite species, Nanhermannia coronata. The manuscript focuses on the ontogenetic development of body and leg chaetotaxy of all developmental stages of the mite and provides abundance data from differents microhabitats. 

This study is important as it clearly separates N. coronata from the closely related species of the genus using combined data, molecular and morphological, not only of adults but also of immatures. Moreover, the ecological part of the study indicates the differences on the ecological preferences of two species with very close morphology (N. coronata και N. sellnicki). 

For the aforementioned reasones I recommend the manuscript for publication in present form.

Sincerely 
